# Enhancing Plaque Segmentation in CCTA with Prompt-based Diffusion Data Augmentation

**Yizhe Ruan**[1,2]**, Xuangeng Chu**[1]**, Ziteng Cui**[1]**, Yusuke Kurose**[1,2]**, Junichi Iho**[3]**,**
**Yoji Tokunaga**[3]**, Makoto Horie**[3]**, Yusaku Hayashi**[3]**, Keisuke Nishizawa**[3]**,**
**Yasushi Koyama**[3,2]**, Tatsuya Harada**[1,2]
[1]**The University of Tokyo**
[2]**RIKEN Center for Advanced Intelligence Project**
[3]**Sakurabashi Watanabe Advanced Healthcare Hospital**
`ruanyizhe@mi.t.u-tokyo.ac.jp`

Reviewed on OpenReview: `https://openreview.net/forum?id=hbTYt8PX9n`

## Abstract

Coronary computed tomography angiography (CCTA) is essential for non-invasive assessment of coronary artery disease (CAD). However, accurate segmentation of atherosclerotic plaques remains challenging due to data scarcity, severe class imbalance, and significant variability between calcified and non-calcified plaques. Inspired by DiffTumor's tumor synthesis and PromptIR's adaptive restoration framework, we introduce PromptLesion [1], a prompt-conditioned diffusion model for multi-class lesion synthesis. Unlike single-class methods, our approach integrates lesion-specific prompts within the diffusion generation process, enhancing diversity and anatomical realism in synthetic data. We validate PromptLesion on a private CCTA dataset and multi-organ tumor segmentation tasks (kidney, liver, pancreas) using public datasets, achieving superior performance compared to baseline methods. Models trained with our prompt-guided synthetic augmentation significantly improve Dice Similarity Coefficient (DSC) scores for both plaque and tumor segmentation. Extensive evaluations and ablation studies confirm the effectiveness of prompt conditioning.

## 1 Introduction

Coronary artery disease (CAD) is a leading cause of mortality worldwide, significantly affecting global death rates Mensah et al. (2023). Coronary computed tomography angiography (CCTA) is widely used as a non-invasive imaging modality for detecting and characterizing coronary plaques, offering a strong alternative to invasive angiography Marano et al. (2020); Serruys et al. (2021). Accurate segmentation of plaque regions in CCTA is crucial for quantifying disease burden and guiding interventions.

However, this task is hampered by high inter-class variability between plaque types (calcified vs. non-calcified plaques) and limited annotated datasets Huang et al. (2020); Lin et al. (2022); Ruan et al. (2025). Moreover, class imbalance is severe since healthy vessel pixels vastly outnumber plaque pixels, which can cause segmentation models to be biased toward normal anatomy.

Recent advances in deep learning-based segmentation have achieved promising results in medical imaging. Convolutional architectures like U-Net Ronneberger et al. (2015) and its self-configuring variant nnU-Net Isensee et al. (2018) have become standard, and transformer-based models (e.g. TransUNet Chen et al. (2021)) further improved multi-organ segmentation by capturing long-range context. Nevertheless, these models require large-scale annotated data to generalize well, making data augmentation essential to alleviate data scarcity. Traditional augmentation techniques (random flips, rotations, etc.) provide limited benefit as

---

[1]`https://github.com/RuanYizhe-77/PromptLesionTMLR`

they do not introduce new anatomical patterns or pathology variations. This limitation motivates the use of generative models to create realistic synthetic lesions and images for augmentation. Generative adversarial networks (GANs) Goodfellow et al. (2014); Armanious et al. (2020); Wu et al. (2024) and variational autoencoders Pesteie et al. (2019); Esser et al. (2021) have been explored for medical data augmentation, but can suffer from mode collapse and limited diversity. Recently, diffusion models have emerged as a powerful class of generative models capable of producing high-fidelity images by iterative denoising. In medical imaging, diffusion models have shown promise in synthesizing anatomically realistic structures. For example, Chen et al. (2024) introduced DiffTumor, a diffusion-based framework for realistic tumor synthesis across organs, demonstrating improved generalization over GANs. However, DiffTumor is restricted to single-class lesion generation, focusing on one tumor type at a time. This limits its applicability for tasks where multiple lesion classes coexist.

Inspired by PromptIR's success Potlapalli et al. (2023) in adaptive image restoration using task-specific prompts, we propose to integrate prompt-based conditioning into a diffusion model for multi-class lesion synthesis. Our method, termed PromptLesion for multi-class lesion generation, incorporates learnable lesion-specific prompts during the diffusion generation process. By conditioning on a prompt that encodes the target lesion type, our model can synthesize both calcified and non-calcified plaques in Coronary Computed Tomography Angiography (CCTA) within a single framework. This controlled augmentation addresses the data imbalance by generating diverse examples of under-represented classes. We extend this approach beyond CCTA plaques to multi-organ tumor synthesis: by conditioning on a lesion-specific prompt (e.g. kidney vs. liver tumor, early vs. late stage), we enable a unified model to generate a variety of tumor types. Our hypothesis is that prompt-guided diffusion augmentation will produce more diverse and realistic lesions, leading to improved segmentation performance on both plaque and tumor segmentation tasks. We validate this by training segmentation models on: (a) real data only, and (b) real + synthetic augmented data, comparing outcomes. Experiments on a private CCTA dataset show that adding synthetic plaques yields notable DSC improvements for plaque segmentation. Similarly, on public kidney Heller et al. (2020), liver Bilic et al. (2023), and pancreas Antonelli et al. (2022) CT datasets, our multi-class synthetic tumors improve segmentation accuracy across all organs, outperforming the DiffTumor augmentation approach.

In summary, our work makes the following contributions:

- **Prompt-conditioned diffusion augmentation:** We introduce PromptLesion, the first diffusion-based framework integrating lesion-specific prompts for controlled multi-class lesion synthesis. This design provides an easy-to-scale solution adaptable to multiple lesion classes and anatomies.

- **Enhanced segmentation via synthetic data:** Our synthetic augmentation method significantly boosts segmentation performance, effectively addressing class imbalance by oversampling rare lesions realistically.

- **Generalization across anatomies:** We validate PromptLesion across CCTA plaque and multi-organ tumor datasets (kidney, liver, pancreas), achieving superior segmentation accuracy and lesion diversity compared to baseline methods.

This paper is structured as follows: Section 2 reviews related work on medical image synthesis, diffusion models, and prompt-based learning. Section 3 details our prompt-conditioned diffusion methodology, including the VQGAN-based autoencoder and multi-class prompt integration. Section 4 presents experiments on CCTA and tumor datasets, with comparisons to baselines and ablations. Section 5 concludes with key findings and future directions.

## 2 Related Work

### 2.1 Medical Image Synthesis and Augmentation

Data Augmentation in Segmentation: Data scarcity and class imbalance are common challenges in medical image segmentation, and various augmentation strategies have been proposed. Beyond basic transformations,

researchers have explored generating synthetic data to enrich training sets. Generative Adversarial Networks (GANs) have been widely used for medical image synthesis and augmentation. For example, Frid-Adar et al. (2018) used GANs to generate synthetic liver lesions in CT scans to improve tumor classification performance. Bowles et al. (2018) applied GAN-based augmentation for brain lesion segmentation, reporting improved segmentation accuracy. In digital pathology, Xue et al. (2021) introduced HistoGAN, a conditional GAN that synthesizes histopathology image patches conditioned on cancer class, yielding significant gains in classification of rare cancer subtypes. These works demonstrate that synthetic lesion data can effectively mitigate class imbalance and improve model generalization in various medical imaging contexts. However, most GAN-based approaches are class-specific – they train separate models for each lesion type or organ. This limits scalability and fails to exploit potential commonalities between classes. Moreover, GANs can struggle to maintain anatomical realism for complex structures. These limitations motivate the use of diffusion models, which offer improved stability and diversity.

## 2.2 Diffusion Models for Medical Image Synthesis

Diffusion probabilistic models have recently shown impressive results in image generation. A diffusion model gradually adds noise to training images and learns to reverse this noising process, enabling it to sample realistic images from pure noise. Ho et al. (2020) introduced the DDPM formulation and demonstrated high-quality image synthesis on natural images. Due to their capacity for generating fine details and mode coverage, diffusion models are well-suited to medical image synthesis, where preserving anatomical fidelity is critical. Several works have adapted diffusion models to medical tasks. Diffusion-based augmentation was applied by Chen et al. (2024)) in DiffTumor, which synthesized tumors on CT and improved cross-organ tumor detection. Their diffusion-generated tumors enabled the training of a segmentation model that generalized to multiple organs, highlighting diffusion models' potential in biomedical data augmentation. Kazerouni et al. (2023) survey diffusion models in medical imaging and note their success in anomaly detection and modality translation tasks. Zhang et al. (2024) propose DiffBoost, a controllable text-guided diffusion model for medical image segmentation augmentation. By incorporating structural edge information and textual labels, DiffBoost generates diverse synthetic images that improve segmentation performance on ultrasound, CT, and MRI datasets. Dorjsembe et al. (2024) introduce Med-DDPM, a 3-D mask-conditioned diffusion model that synthesises anatomically faithful brain-MRI volumes from tumour masks, further underscoring conditional diffusion's value for privacy-preserving data augmentation. These results reinforce that diffusion-based synthetic data can significantly boost downstream segmentation accuracy. Compared to GANs, diffusion models offer improved mode diversity and training stability at the cost of higher computation. Our work builds upon these advances by introducing conditional diffusion for multiple lesion classes simultaneously. While diffusion models have been successfully applied to tasks such as anomaly detection and single-class lesion synthesis (e.g., for a specific tumor type), many recent works also explore domain translation, denoising, reconstruction, and multi-class tasks. Nonetheless, the majority of lesion-focused diffusion studies to date often concentrate on one particular lesion type rather than simultaneously addressing multiple classes; we extend the paradigm to multi-class lesion generation via prompt embeddings, as described next.

## 2.3 Prompt-based Learning in Generative Models

Prompt learning – providing additional conditioning inputs or context to guide model behavior – has gained traction in both NLP and vision Brown et al. (2020); Liu et al. (2023); Shin et al. (2020); Radford et al. (2021); Zhou et al. (2022); Jia et al. (2022). In NLP, prompt templates are used to steer large language models to perform specific tasks without fine-tuning. In computer vision, learnable prompts or tokens have been employed to adapt transformers to new tasks (e.g. LoCoOP Miyai et al. (2023) for few-shot classification). Prompt-based techniques have only recently been explored in image generation and restoration. Potlapalli et al. (2023) introduced PromptIR, which uses trainable prompt vectors to encode degradation information (noise, blur, haze levels) for image restoration. By injecting prompts into a restoration network, PromptIR achieved state-of-the-art results on multiple degradation types with a single model. This demonstrated that lightweight prompts can effectively modulate a deep network's behavior based on task-specific context. In

the generative domain, approaches like text-to-image diffusion Zhang et al. (2023) allow controlling outputs via textual prompts. Yet, prompt-based conditioning in medical generative models remains underexplored.

PromptIR's plug-and-play prompt modules inspire our work: we design a prompt-based conditioning mechanism for a diffusion generator. The prompts in our model encode lesion type information (plaque or tumor category) and guide the image synthesis process to produce the desired abnormality. This is conceptually related to conditional generation but with a flexible learned prompt vector instead of a simple one-hot label or fixed scalar conditioning. The prompt is injected at multiple stages of the diffusion U-Net, allowing it to influence the denoising trajectory towards the target lesion appearance. We show that such prompt conditioning yields distinct benefits in a multi-class setting, as it provides a form of adaptive guidance that can be tuned for each lesion class without retraining the entire model.

### 2.4 Applications of Multi-Class Lesion Synthesis

While single-class synthetic augmentation has demonstrated benefits in medical imaging contexts, such as liver lesions Frid-Adar et al. (2018), brain lesions Bowles et al. (2018), and pathology classification Xue et al. (2021), the capability to simultaneously handle multiple lesion classes within a single framework is becoming increasingly crucial. In clinical practice, multiple lesion types or pathologies frequently coexist within a single imaging scan, such as varying plaque compositions in coronary arteries Ruan et al. (2025) or metastatic tumors distributed across multiple organs. Nevertheless, prior generative augmentation methods, including GAN-based techniques Armanious et al. (2020); Wu et al. (2024) and diffusion-based methods like DiffTumor Chen et al. (2024), typically focus on synthesizing a single lesion class per model. This single-class strategy does not effectively exploit shared anatomical or pathological characteristics, leading to inefficiencies and limited scalability when expanding to multiple lesion types.

Our work addresses this limitation by introducing a unified, prompt-driven diffusion framework designed explicitly for multi-class lesion synthesis. We target two distinct clinical domains: coronary artery plaques (calcified and non-calcified) and abdominal organ tumors (kidney, liver, pancreas, each with early and late stages). In the context of coronary computed tomography angiography (CCTA), our method synthesizes both plaque types within the arterial context, providing balanced lesion diversity that enhances segmentation robustness on scans with mixed plaque compositions. Similarly, for tumor segmentation tasks, our approach generates diverse tumors across different abdominal organs, allowing segmentation models to learn generalized lesion features rather than becoming overly specialized on a single organ or lesion class.

Leveraging prompt-conditioned diffusion ensures anatomical coherence in synthesized images despite variations in lesion appearance. Consequently, we produce richly augmented datasets that encompass multiple lesion categories, significantly improving segmentation accuracy across lesion classes. Our experimental results consistently demonstrate superior Dice similarity coefficients for both plaque and tumor segmentation tasks compared to baseline methods trained solely on real data or single-class augmentation strategies. Overall, prompt-based multi-class lesion synthesis provides a scalable, generalizable, and effective augmentation strategy, representing a significant advancement in generative augmentation for medical imaging.

## 3 Method

Our proposed method combines a latent diffusion model with prompt-based multi-class conditioning to generate realistic lesions for data augmentation. The overall pipeline (illustrated in Figure 1) consists of three stages: (1) a VQGAN autoencoder that learns a compact latent representation of medical images, (2) a conditional diffusion model operating in the latent space, guided by lesion prompts to produce synthetic abnormalities, and (3) a segmentation model to evaluate the performance of synthetic images. We describe each component and its integration below.

### 3.1 VQGAN-based Reconstruction

Following the approach introduced by DiffTumor Chen et al. (2024), we employ a Vector Quantized Generative Adversarial Network (VQGAN) Esser et al. (2021) to encode medical images into a compact latent

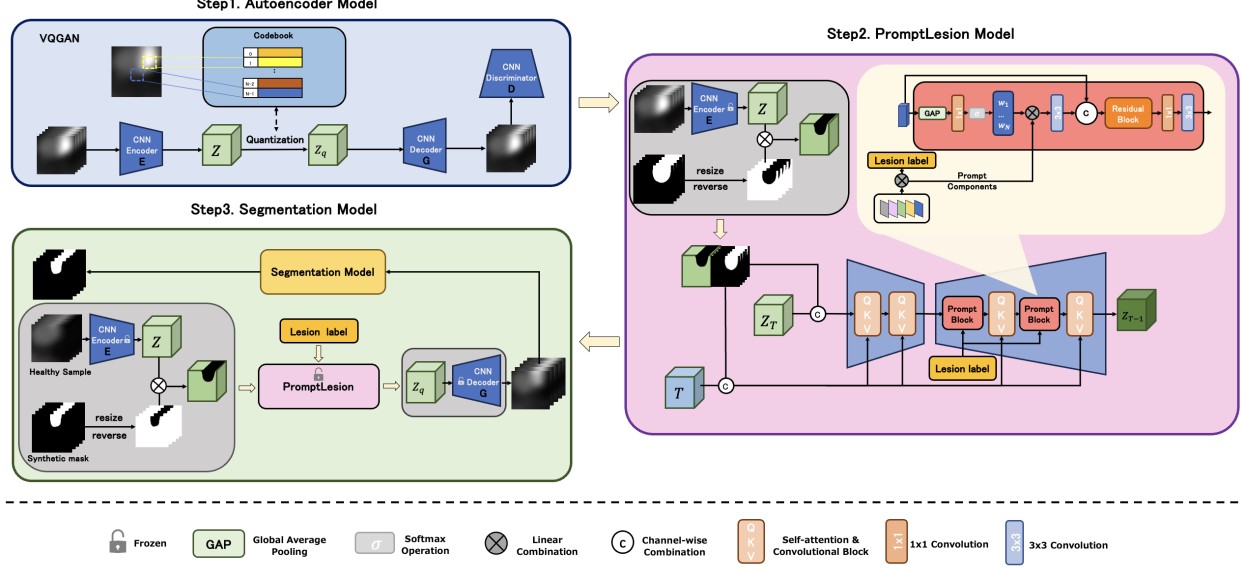

Figure 1: Overview of our Prompt-based Diffusion augmentation framework. (a) Step 1: VQGAN Autoencoder encodes input medical images into compact latent representations. (b) Step 2: Prompt-conditioned diffusion model named PromptLesion synthesizes lesion-specific latent codes guided by prompt embeddings. (c) Step 3: Segmentation Model evaluates the performance of synthetic lesion images. See Section 3 for detailed descriptions.

space, as step 1 in Figure 1 shows. Working in this latent representation significantly reduces computational demands and enables the diffusion model to focus better on meaningful anatomical and pathological structures.

Specifically, given an input 3D medical image patch $x \in \mathbb{R}^{H \times W \times D}$, the encoder $E_\phi$ produces a compressed latent representation $z = E_\phi(x) \in \mathbb{R}^{h \times w \times d}$. Each element of $z$ is subsequently quantized by mapping it to the nearest vector from a learned discrete codebook $\mathcal{Z} = \{e_i\}_{i=1}^{K}$, yielding the quantized latent code $z$. The decoder $G_\phi$ then reconstructs the image from this quantized code, resulting in the reconstructed output $\tilde{x} = G_\phi(z)$.

To train VQGAN effectively, we optimize a combined loss objective comprising several terms: a pixel-level $L_1$ reconstruction loss ($\mathcal{L}_{rec}$) ensuring accurate overall reconstruction, a perceptual loss ($\mathcal{L}_{perceptual}$) to preserve anatomical details across axial, coronal, and sagittal planes, and two codebook-related losses—a codebook optimization loss ($\mathcal{L}_{codebook}$) and an encoder commitment loss ($\mathcal{L}_{commit}$)—which collectively stabilize quantization training. Additionally, we introduce adversarial training with two discriminators: a volumetric discriminator ($D_v$) enforcing overall 3D anatomical coherence and a slice-based discriminator ($D_s$) improving slice-wise realism. We further incorporate a discriminator-based feature matching loss ($\mathcal{L}_{match}$) to guide the decoder toward generating realistic intermediate representations. The discriminator loss ($\mathcal{L}_{disc}$) is thus formulated as:

$$\mathcal{L}_{disc} = \log D_{s/v}(x) + \log \left(1 - D_{s/v}(\tilde{x})\right). \tag{1}$$

After training, encoder and decoder parameters are frozen. The diffusion model operates exclusively on this discrete latent space, ensuring efficient synthesis while retaining critical anatomical information.

## 3.2 Diffusion Model with Prompt Conditioning

Inspired by the approach outlined in DiffTumor Chen et al. (2024), we employ a Denoising Diffusion Probabilistic Model (DDPM) to synthesize latent representations containing realistic lesions, as step 2 in Figure 1 shows. Specifically, we define a diffusion model using a U-Net architecture $f_\theta(z_t, t, m, z^{healthy}, p)$ designed

to reverse a predefined forward noising process. Note that, similarly to DiffTumor, the diffusion model is conditioned on both the lesion mask $m$ and the healthy region $z^{healthy} = (1 - m) \odot z$. During the forward process, Gaussian noise is progressively introduced into the initial latent representation $z_0$ (obtained from real images via the trained VQGAN encoder), resulting in a noisy latent code $z_t$ at timestep $t \in [1, T]$. The U-Net then predicts the added noise given $z_t$, timestep $t$, lesion mask $m$, healthy CT region $z^{healthy}$, and a lesion-specific prompt embedding $p$.

We explicitly introduce prompt conditioning to guide lesion synthesis, integrating prompts at multiple scales within the U-Net architecture. Prompts encode lesion class labels(one-hot) into dense vectors using a prompt embedding network (e.g. $[0, 1, 0]$ for calcified plaque, $[0, 0, 1]$ for non-calcified plaque in CCTA datasets; $[0, 1, 0, 0, 0, 0, 0]$ for early stage tumor in liver datasets). Our framework further supports controlled multi-class lesion synthesis via user-specified prompts (Section3.2.1) and leverages adaptive prompt learning through implicit prompts derived from a transformer-based encoder to enrich contextual relationships among lesion types (Section3.2.2). Formally, the diffusion model is trained by optimizing the standard DDPM objective, minimizing:

$$\mathbb{E}z_0, c, \epsilon, t \left[ |\epsilon - f\theta(z_t, t, m, z^{healthy}, p)|^2 \right], \tag{2}$$

where $f\theta(z_t, t, m, z^{healthy}, p)$ is our PromptLesion Network, as a 3D U-Net variant with prompt block, interleaved self-attention layers and convolutional layers Ho et al. (2020); Nichol & Dhariwal (2021); the noisy latent $z_t$ is obtained through the forward noising step defined by:

$$z_t = \sqrt{\bar{\alpha}_t} z_0 + \sqrt{1 - \bar{\alpha}_t} \epsilon, \quad \epsilon \sim \mathcal{N}(0, I). \tag{3}$$

Through this training, the U-Net learns to progressively denoise latents towards plausible, prompt-specific anatomical structures and lesions.

### 3.2.1 Multi-class Lesion Generation via Prompts

Once trained, our diffusion model enables controlled synthesis of lesions conditioned on user-specified prompts. For instance, to generate synthetic coronary plaques, we first sample a random Gaussian latent $z_T \sim \mathcal{N}(0, I)$ and select a desired lesion class $c_{\text{plaque}} \in$ calcified, non-calcified . The prompt embedding network generates the corresponding prompt vector $p$. We then perform iterative reverse denoising steps, computing:

$$z_{t-1} = f_\theta(z_t, t, m, z^{healthy}, p), \quad t = T, T - 1, \ldots, 1, \tag{4}$$

ultimately yielding a clean latent $z_0$ representative of a vessel containing the desired plaque type. Decoding $z_0$ via the pretrained VQGAN decoder produces synthetic but anatomically coherent CCTA images suitable for segmentation training.

A similar procedure is applied for tumor data, where prompts condition the model to generate organ-specific and stage-specific tumor lesions. Crucially, our unified diffusion model supports generation of multiple lesion types using a single architecture, differentiating lesion classes solely through prompt vectors. The model can also generate lesion-free images by employing a dedicated "healthy" identity prompt, enabling versatile data augmentation strategies, including guided insertion of synthetic lesions into real lesion-free backgrounds through partial reverse diffusion steps analogous to latent-space inpainting.

Following DiffTumor Chen et al. (2024), we also adopt lesion-background blending, which they showed is superior to fully synthetic volumes. Our augmentation procedure integrates synthetic lesion patches into real images at anatomically plausible locations guided by clinical domain knowledge, such as inserting synthetic plaques along coronary artery walls.

### 3.2.2 Adaptive Prompt Learning

To further enhance the informativeness and representational capability of our prompt-based augmentation framework, we extend the prompt encoding mechanism beyond simple one-hot vectors by incorporating implicit prompts extracted through a lightweight transformer-based encoder, as shown in Figure 2. Specifically, instead of directly using a one-hot encoded vector of lesion labels as explicit conditioning input, we employ

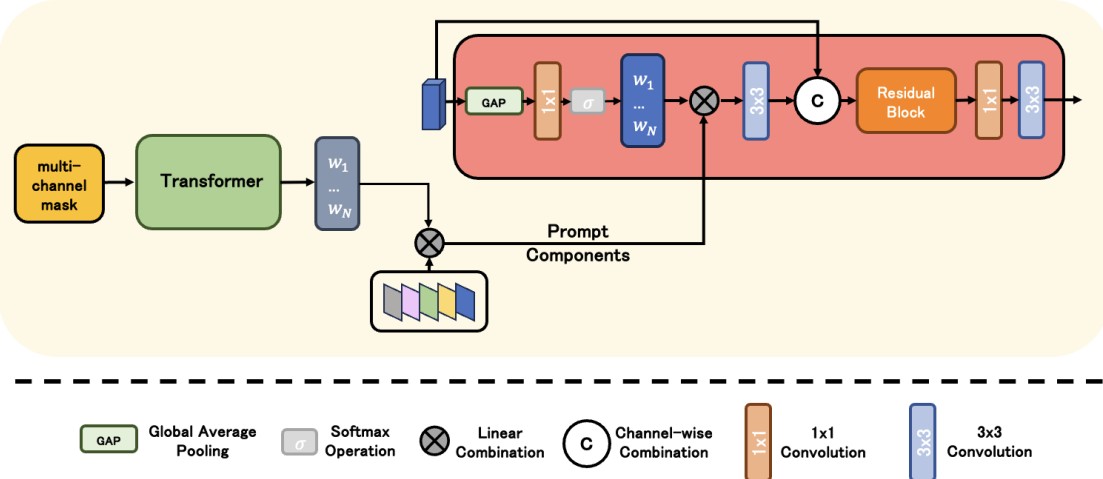

Figure 2: Transformer-based Prompt Network: Instead of conditioning with one-hot vector, transformer-based prompt network conditions with multi-channel mask.

a small transformer encoder Vaswani et al. (2017) to embed the multi-class segmentation masks into latent embeddings. These embeddings inherently capture richer contextual relationships among different lesion classes, thus serving as more informative implicit prompts to guide the diffusion process. The lightweight transformer encoder consists of 3 layers, each comprising multi-head self-attention and position-wise feed-forward networks, effectively balancing representational capacity and computational efficiency. Compared to the explicit one-hot encoding used previously, this implicit prompting strategy leverages spatial and class relational context, potentially improving the generation fidelity and diversity of synthetic lesions, as demonstrated empirically in subsequent experiments.

### 3.3 Segmentation Model

As shown in step 3 of Figure 1, to evaluate our synthetic augmentation, we use standard segmentation models including 3D U-Net and nnU-Net, following the general methodology of DiffTumor Chen et al. (2024). Specifically, for the CCTA plaque segmentation task, we combine synthetic plaque-containing slices generated by PromptLesion with healthy artery slices from the real training dataset. Each synthetic slice includes automatically generated lesion masks indicating calcified or non-calcified plaques, thus enriching rare lesion classes while maintaining realistic artery anatomy.

For the multi-organ tumor segmentation tasks (kidney, liver, pancreas), we directly follow DiffTumor's procedure, integrating PromptLesion-generated synthetic tumor images and their corresponding masks with real tumor data. These segmentation models serve as evaluation tools to measure the effectiveness of our synthetic data—by comparing models trained with and without PromptLesion augmentation, we assess whether the generated images contribute to improved performance on downstream segmentation benchmarks.

## 4 Experiments

### 4.1 Datasets

Our method was trained and evaluated on two primary tasks: plaque segmentation in CCTA and tumor segmentation in abdominal CT.

**Private CCTA Dataset.** For plaque-related experiments, we used a private set of 100 cardiac CTA volumes acquired at the Sakurabashi Watanabe Advanced Healthcare Hospital. Following the data split

ratio of a prior work Ruan et al. (2025), the cohort was divided into a training set of 70 patients, a validation set of 10, and a test set of 20. The annotation procedure was carried out by cardiac experts. For each CCTA volume, the experts provided detailed, slice-by-slice segmentation masks using specialized cardiac analysis software. These manual annotations delineated five distinct classes: background, artery lumen, artery wall, calcified plaque, and non-calcified plaque. To ensure the quality and consistency of the ground truth, the annotation process involved a multi-stage review where each segmentation was cross-verified by a second senior expert. This study was conducted with IRB approval granted by the Ethics Committee of RIKEN (No. Wako2022-14), and all data was fully anonymized.

**Public Tumor Datasets.** For tumor segmentation, we used the same public datasets as DiffTumor Chen et al. (2024): the Kidney Tumor dataset (KiTS) Heller et al. (2020), the Liver Tumor dataset (LiTS) Bilic et al. (2023), and the Pancreas Tumor dataset (MSD Pancreas) Antonelli et al. (2022). We combined their training sets to form a multi-organ tumor training set and evaluated performance on each organ's standard test split.

**Pre-training Dataset for VQGAN.** Following the procedure in DiffTumor Chen et al. (2024), the VQGAN autoencoder for the tumor task was pre-trained on the AbdomenAtlas-8K dataset Qu et al. (2023). For the plaque task, the VQGAN was pre-trained on our private CCTA training set.

## 4.2 Implementation Details

**Data Curation and Preprocessing.** For CT preprocessing, we followed the procedure used in DiffTumor Chen et al. (2024), extracting fixed-size 3D sub-volumes from the original CT volumes for training the diffusion model. For CCTA preprocessing, the raw CCTA data was provided as multi-slice TIFF files. We first developed a pipeline to convert this data into the Nifti format. For each patient's artery, TIFF slices were stacked to form a 3D volume. The original Hounsfield Unit (HU) values were scaled to grayscale intensity values in the range [0, 255]. The corresponding label files were converted into multi-class segmentation masks, where specific grayscale values were mapped to integer class labels (e.g., background: 0, lumen: 1, artery wall: 2, calcified plaque: 3, non-calcified plaque: 4).

This curated Nifti dataset then served as the input to our MONAI-based training pipeline. The key preprocessing and augmentation steps include: (1) reorienting volumes to RAS coordinate system; (2) resampling volumes to a uniform voxel spacing of [**e.g., 1.0 x 1.0 x 1.0 mm³**]; (3) normalizing intensity values from a window of [**a_min, a_max**] to [**b_min, b_max**]; (4) performing class-balanced random cropping to a fixed patch size of $96 \times 96 \times 96$ voxels; and (5) applying random 90-degree rotations.

**Training.** The random seed for all experiments was set to 1234 to ensure reproducibility. Our training pipeline is implemented using the MONAI framework. To address class imbalance, we employed MONAI's class-aware cropping methods, specifically `RandCropByPosNegLabeld` and `RandCropByLabelClassesd`, to ensure that training patches were sampled with a balanced representation of different lesion types. For data augmentation, we applied only random 90-degree rotations (`RandRotate90d` with a probability of 0.1). No other complex augmentations such as affine transformations or elastic deformations were used. Further details on hyperparameters and training schedules are provided in the appendix.

## 4.3 Benchmarking Protocol for CCTA

Due to patient privacy regulations, our CCTA dataset cannot be made publicly available. To facilitate fair comparisons for future research, we propose the following protocol:

- **File Formats:** All inputs and outputs should be in the Nifti (`.nii.gz`) format.

- **Evaluation Metrics:** The primary metric is the Dice Similarity Coefficient (DSC). Performance should be reported as the average DSC for each plaque class individually and as a mean across all classes.

- **Standard Preprocessing:** External methods should, at a minimum, implement standard preprocessing steps, including isotropic resampling and intensity normalization, as detailed in our public code.

## 4.4 Experimental Setup and Baselines

We integrate our synthetic data into the training of segmentation models and compare against multiple baselines:

- Baseline segmentation (Real-Only): Train segmentation networks using only the real training images and labels, with no synthetic augmentation, using baseline U-Net and nnU-Net architectures. This represents the standard supervised scenario with limited data.

- PromptLesion Augmentation (Ours): Train the same segmentation networks on the combination of real and PromptLesion-generated images. For each real image in a batch, we add one or more synthetic images of under-represented classes (e.g. a synthetic non-calcified plaque image if that class is rare) generated by our model. The synthetic images come with automatically generated segmentation masks (since we know the lesion ground-truth in the synthetic world).

- PromptLesion[†] Augmentation (Ours): In the additional experiment, we thoroughly evaluated the impact of using transformer-derived implicit prompts compared to the conventional one-hot vector approach. We conducted training of the PromptTumor model using the implicit transformer-based prompts under identical conditions as previous experiments. Segmentation results were evaluated on the same tumor datasets (liver, kidney, pancreas) and coronary artery plaque datasets, using baseline U-Net and nnU-Net architectures.

- DiffTumor Augmentation: As an additional baseline, we simulate using DiffTumor for augmentation. DiffTumor can only synthesize one type of tumor at a time (e.g. liver tumors). We generate synthetic tumors for each organ separately using DiffTumor's released model and add those to training. This baseline tests if our multi-class approach yields an advantage over using multiple single-class generators.

- SynTumor Augmentation: We compare our diffusion-based augmentation with SynTumor Hu et al. (2023). These images augment the training set similarly for both CCTA and tumor datasets.

- Med-DDPM (multi-class) Augmentation: We extend Med-DDPM Dorjsembe et al. (2024) to handle all lesion classes jointly by conditioning on multi-class masks. For each healthy CT, a lesion mask of a target class is sampled; Med-DDPM then generates a lesion volume that is composited into the CT following the DiffTumor blending strategy. The resulting synthetic scans and their masks are mixed with the real data using the same sampling ratio as PromptLesion, allowing us to assess a mask-conditioned diffusion baseline without prompt embeddings.

- Segmentation architectures: We report results with two segmentation models: a standard 3D U-Net and the nnU-Net 3D framework. We train each model on each training set variant (Real-Only-Training, Mix-Training, etc). All models are trained for 500 epochs with early stopping on the validation set.

Note that we use the DiceCELoss Sudre et al. (2017) for all the segmentation trainings. We emphasize that the synthetic images are used in addition to real images (not replacing them). The proportion of synthetic data is tuned on the validation set – we found using a 1:1 ratio of synthetic: real achieved the best results without overwhelming the network with possibly redundant synthetic samples.

## 4.5 Evaluation Metrics and Protocol

Segmentation performance is evaluated primarily by the Dice Similarity Coefficient (DSC), which measures the overlap between predicted masks and ground truth . We report DSC for each class (calcified vs non-calcified plaque, or tumor in each organ) and overall mean DSC. We compare model performance on the

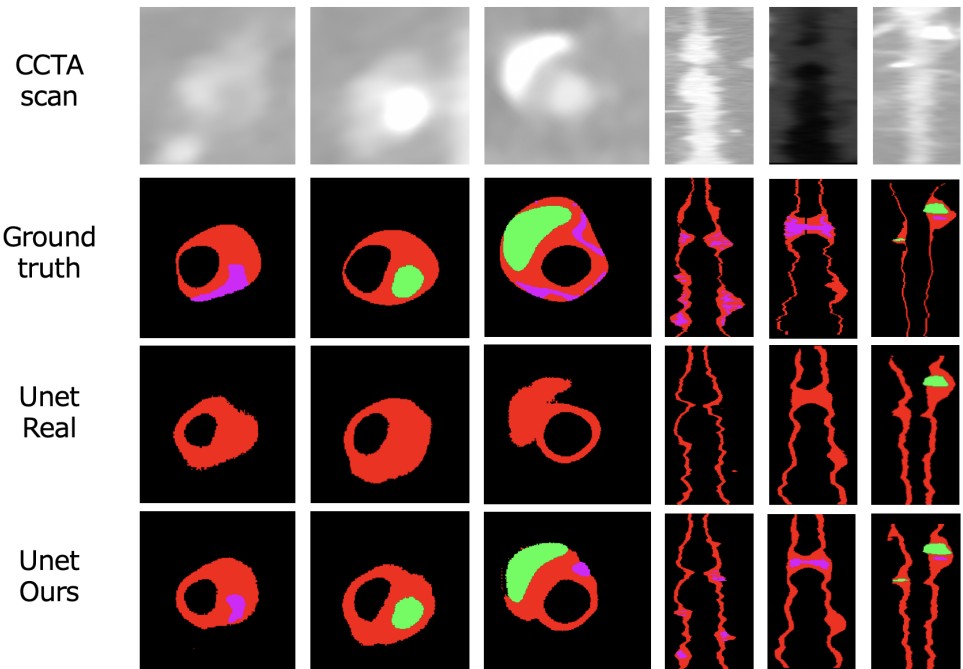

Figure 3: Vision Comparison for CCTA datasets: In mask images, black refers to background, red denotes coronary artery wall, green indicates calcified plaque, and purple is non-calcified plaque

independent test sets for each task:(a) CCTA Plaque Segmentation: We evaluate on 10 held-out CCTA volumes, computing the per-lesion DSC. (b) Multi-organ Tumor Segmentation: We test on the official test sets of KiTS, LiTS, and MSD Pancreas. Following DiffTumor, we perform evaluation tumor-wise and report the mean DSC across all tumors.

For statistical significance, we run each experiment 3 times with different random seeds and report the average DSC.

### 4.6 Results

Figure 3 shows qualitative segmentation results on the CCTA dataset, comparing a baseline model (trained on real data only) with our proposed PromptLesion. Our method effectively captures challenging plaque regions, particularly non-calcified plaques, which often have ambiguous boundaries and relatively low contrast with surrounding tissues. In contrast, the baseline segmentation model often misses these subtle plaque structures or incorrectly segments partial regions, leading to incomplete or noisy segmentations. By leveraging our prompt-conditioned diffusion augmentation, the segmentation predictions become more accurate and closely aligned with expert annotations, demonstrating enhanced sensitivity towards subtle lesions.

Similarly, Figure 4 illustrates visual comparisons on the tumor datasets (kidney, liver, pancreas). The model trained with our prompt-based diffusion augmentation produces more accurate tumor masks across organs compared to the baseline, especially for smaller or less conspicuous lesions. Figure 4 demonstrates similarly enhanced segmentation accuracy for kidney, liver, and pancreas tumors, highlighting our method's generalizability.

Quantitatively, Table 1 summarizes DSC results for CCTA plaque segmentation. Our PromptLesion and PromptLesion[†] methods consistently outperformed the baselines, achieving marked improvements particularly for the challenging non-calcified plaques. PromptLesion[†] provided the best overall performance across both segmentation architectures.

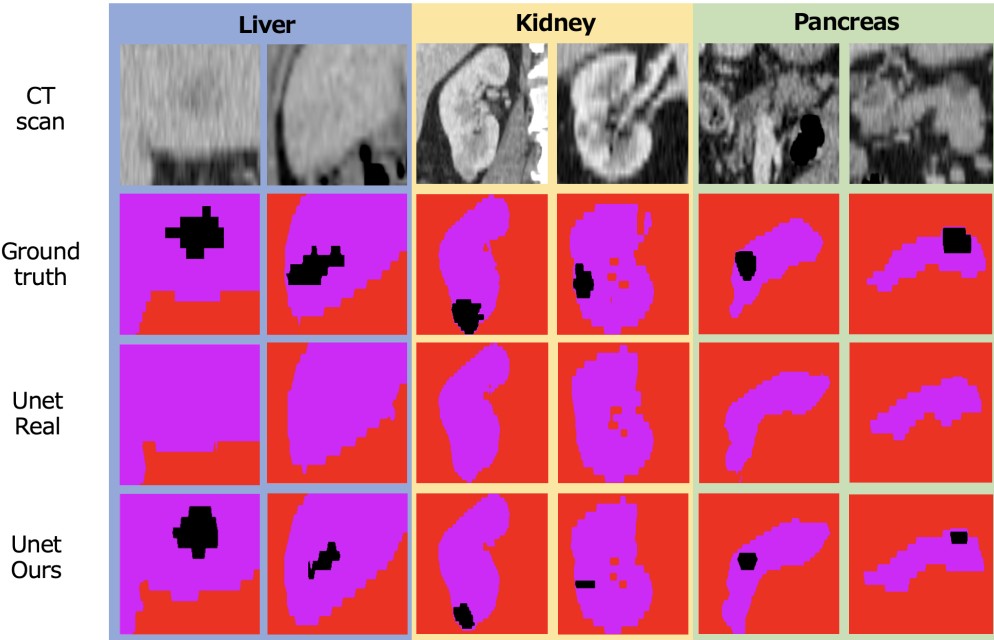

Figure 4: Vision Comparison for Tumor Datasets: In mask images, red is the background, purple refers to organ, and the black is the target tumor.

Table 2 compares the segmentation performance of various approaches for multi-class tumor segmentation, evaluated on three different organs (liver, kidney, pancreas) with two segmentation backbones (U-Net and nnU-Net). "Real Tumors" represents training on real tumor data only (no synthetic augmentation). "DiffTumor$_{avg}$" are baselines that use single-source training (trained on one organ's data) but are tested across multiple domains (performance averaged across organs). Our proposed methods, "PromptLesion" and "PromptLesion†," consistently outperform these baselines (including DiffTumor$_{avg}$) in multi-domain segmentation without sacrificing performance. The results demonstrate that PromptTumor achieves the best overall performance (see the "Avg" column), highlighting its effectiveness in generating multi-class tumors across different domains.

### 4.7 Ablation Study

#### 4.7.1 Multi-class tumor generation across domains

Table3 presents an ablation study evaluating multi-class tumor generation across three domains (liver, kidney, pancreas) using two different backbones (UNet and nnUNet). "DiffTumor" denotes the baseline method trained on the same source data, while "PromptLesion" (and its variant "PromptLesion†") represents our proposed approach for multi-class generation. The results indicate that PromptLesion achieves comparable or even higher performance than DiffTumor, demonstrating that our prompt-based framework does not sacrifice performance when extending tumor generation to multiple classes.

#### 4.7.2 Prompt conditioning

To systematically investigate the contribution of transformer-based implicit prompts, we performed an ablation study by comparing: (a)Baseline (Real Data Only): Training segmentation models solely on real data.(b)DiffTumor : Synthetic augmentation using the DiffTumor method. (c) PromptLesion (One-hot): Our initial PromptTumor method conditioned on one-hot vectors of target lesions. (d)PromptLesion† (Transformer-based Implicit Prompt): The improved method introducing latent embeddings generated from multi-class masks through a lightweight transformer encoder. We observed a consistent performance boost when transitioning from explicit one-hot prompts to implicit prompts, highlighting that implicit embeddings

Table 1: Quantitative comparison of plaque generation in Dice Score.

| Backbone | Method | Calcified Plaque | Non-calcified Plaque | Avg |
|---|---|---|---|---|
| Unet | Real CCTA | 68.1 | 21.3 | 44.7 |
| | SynTumor | 69.5 | 27.2 | 48.4 |
| | DiffTumor$_{avg}$ | 70.2 | 30.2 | 50.2 |
| | Med-DDPM | 71.6 | 34.3 | 52.9 |
| | PromptLesion | 73.4 | 38.2 | 55.8 |
| | PromptLesion$^{\dagger}$ | **75.1** | **39.0** | **57.1** |
| nnUnet | Real CCTA | 69.2 | 23.1 | 46.2 |
| | SynTumor | 69.9 | 28.3 | 49.1 |
| | DiffTumor$_{avg}$ | 70.6 | 31.7 | 51.1 |
| | Med-DDPM | 70.9 | 35.3 | 53.1 |
| | PromptLesion | 74.1 | 40.1 | 57.1 |
| | PromptLesion$^{\dagger}$ | **75.6** | **44.3** | **59.9** |

Table 2: Quantitative comparison of tumor generation in Dice Score.

| Backbone | Method | Liver | Kidney | Pancreas | Avg |
|---|---|---|---|---|---|
| Unet | Real Tumors | 35.8 | 50.0 | 43.5 | 43.1 |
| | SynTumor | 48.3 | 53.3 | 44.2 | 48.6 |
| | DiffTumor$_{avg}$ | 52.3 | 54.2 | 46.3 | 50.9 |
| | Med-DDPM | 59.2 | 54.3 | 52.6 | 55.4 |
| | PromptLesion | 58.6 | 58.9 | 52.9 | 56.8 |
| | PromptLesion$^{\dagger}$ | **64.0** | **60.0** | **53.7** | **59.2** |
| nnUnet | Real Tumors | 47.2 | 51.3 | 33.4 | 43.9 |
| | SynTumor | 49.2 | 54.1 | 36.4 | 46.5 |
| | DiffTumor$_{avg}$ | 53.8 | 55.7 | 39.8 | 49.7 |
| | Med-DDPM | 53.7 | 55.3 | 45.3 | 51.4 |
| | PromptLesion | 56.0 | 59.3 | 45.2 | 53.5 |
| | PromptLesion$^{\dagger}$ | **58.2** | **61.1** | **46.6** | **55.3** |

encode additional structural and semantic information beneficial for the diffusion model. The quantitative evidence in Tables 1 and 2 corroborates this observation, revealing an average DSC gain of approximately 2-3 percentage points on both plaque and tumor segmentation benchmarks relative to the original PromptLesion.

Table 3: Ablation study for multi-class tumor generation across domains

| Backbone | Method | liver | kidney | pancreas | Avg |
|---|---|---|---|---|---|
| Unet | DiffTumor | 59.3 | **60.9** | **54.2** | 58.1 |
| | PromptLesion | 58.6 | 58.9 | 52.9 | 56.8 |
| | PromptLesion$^\dagger$ | **64.0** | 60.0 | 53.7 | **59.2** |
| nnUnet | DiffTumor | **58.9** | **62.8** | 44.8 | **55.5** |
| | PromptLesion | 56.0 | 59.3 | 45.2 | 53.5 |
| | PromptLesion$^\dagger$ | 58.2 | 61.1 | **46.6** | 55.3 |

## 5 Conclusion

We proposed **PromptLesion**, a prompt-conditioned latent-diffusion framework that tackles data scarcity and severe class imbalance in coronary computed tomography angiography (CCTA). By injecting lesion-specific prompts into the denoising trajectory, the model synthesizes anatomically coherent calcified and non-calcified plaques, supplying segmentation networks with hard-to-obtain training samples. Across both backbone architectures, the augmented models consistently outperform the real-data baselines in plaque Dice score, with especially pronounced improvements for the under-represented non-calcified plaques.

To probe generalization, we reused the same framework—changing only the prompt definitions—on public multi-organ tumor datasets. Consistent Dice improvements on liver, kidney, and pancreas benchmarks confirm that prompt-guided diffusion scales gracefully to unseen anatomies and lesion types without retraining the generator.

**Impact:** This work advances the state-of-the-art in generative medical image augmentation by introducing an easily scalable, multi-class synthesis paradigm. Clinically, enhanced plaque segmentation can improve patient risk assessment in coronary artery disease, and improved tumor segmentation has significant potential for better treatment planning. Our prompt-conditioned diffusion framework can be readily adapted to other medical imaging scenarios by defining suitable prompts, underscoring its broad applicability and potential to reduce annotation burdens in clinical AI deployments.

## Broader Impact and Clinical Considerations

Our PromptLesion method addresses the challenges of label scarcity and class imbalance by generating realistic synthetic lesions. We outline the following considerations for its use:

- **Intended Use:** This technology is designed exclusively as a **training-time tool** to enhance model robustness, especially for rare classes. It is **not intended for direct diagnostic use**, and its synthetic outputs should not replace real clinical data.

- **Clinical Risk:** A potential risk involves synthetic lesions containing artifacts that could mislead models or clinicians. Consequently, rigorous and comprehensive validation is essential before any clinical deployment of models trained with this method.

- **Data Privacy:** The clinical data used in this work was fully anonymized under IRB approval. We strongly encourage researchers to apply the same strict privacy standards to their own data.

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
