# Appendix: Enhancing Plaque Segmentation in CCTA with Prompt-based Diffusion Data Augmentation

**Yizhe Ruan**[1,2]**, Xuangeng Chu**[1]**, Ziteng Cui**[1]**, Yusuke Kurose**[1,2]**, Junichi Iho**[3]**,**
**Yoji Tokunaga**[3]**, Makoto Horie**[3]**, Yusaku Hayashi**[3]**, Keisuke Nishizawa**[3]**,**
**Yasushi Koyama**[3,2]**, Tatsuya Harada**[1,2]
[1]**The University of Tokyo**
[2]**RIKEN Center for Advanced Intelligence Project**
[3]**Sakurabashi Watanabe Advanced Healthcare Hospital**
ruanyizhe@mi.t.u-tokyo.ac.jp

**Reviewed on OpenReview:** `https://openreview.net/forum?id=hbTYt8PX9n`

## 1 Training Details

We adopt a three-stage training procedure, similar to the setup in DiffTumor Chen et al. (2024), but extended with our PromptLesion conditioning. Each stage has a distinct network and set of hyperparameters:

### 1.1 Step 1: VQGAN Autoencoder

- **Objective:** Learn a compact latent representation of 3D volumes and reconstruct them with high fidelity.

- **Network Architecture:** A 3D convolutional encoder and decoder, plus a vector quantization module (codebook size = 16384; codebook dim = 8). We also employ one 3D discriminator and one 2D slice-based discriminator.

- **Training Set:** We use AbdomenAtlas-8K Qu et al. (2023) dataset for training. Cropped volumetric patches of size $96 \times 96 \times 96$, covering both healthy and lesion regions. We randomly sample patches from each volumetric scan.

- **Hyperparameters:**
  - Epochs: 1000
  - Batch size: 4
  - Learning rate: $3 \times 10^{-4}$
  - Optimizer: Adam with $\beta_1 = 0.9$, $\beta_2 = 0.999$
  - Loss Terms: $\ell_1$ for reconstruction, codebook commitment loss, adversarial losses (3D/2D discriminators), and perceptual loss.

### 1.2 Step 2: PromptLesion Model

- **Objective:** Synthesize diverse lesions within the latent space. We train a 3D diffusion model, conditioning on healthy background, lesion masks, and prompt embeddings.

- **Diffusion Forward Process:** Gradually add Gaussian noise to the latent representation $z_0$ over $T = 400$ timesteps.

- **Network Architecture:** A 3D U-Net with interleaved self-attention layers. We integrate the lesion-type prompt either via a learnable embedding or a lightweight transformer, alongside the latent healthy region and the binary mask.

- **Hyperparameters:**
  - Train Steps: 60000
  - Batch size: 4
  - Learning rate: $1 \times 10^{-4}$
  - Optimizer: Adam with $\beta_1 = 0.9$, $\beta_2 = 0.999$
  - Noise Schedule: linear
  - Loss Function: Mean-squared error between predicted noise and true noise at each timestep (denoising diffusion objective).

- **Inference Details:** For lesion synthesis, we start from a healthy latent region, then perform reverse diffusion for 400 steps. The prompt embedding selects the lesion type to be generated (e.g., calcified plaque, non-calcified plaque, or tumors in different organs) while the mask inputs decide the position of the lesion.

## 1.3 Step 3: Segmentation Model

- **Objective:** Use both real and synthetic lesion data to train a 3D segmentation network.

- **Training Data Construction:**
  - Real data: Retain original CT volumes with expert annotations.
  - Synthetic data: Convert each latent sample from Step 2 back to voxel space via the pretrained decoder. The diffusion mask serves as the lesion label. We mix real and synthetic samples with a 1:1 ratio in most experiments.

- **Early Stop:**
  - **Early stopping (segmentation only):** stop if validation Dice shows no improvement for 25 consecutive epochs; keep the best-Dice checkpoint for testing.

- **Network Architecture:** We adopt nnU-Net or a 3D U-Net backbone for volumetric segmentation.

- **Hyperparameters:**
  - Epochs: 500
  - Batch size: 12
  - Learning rate: $2 \times 10^{-4}$
  - Optimizer: Adam with $\beta_1 = 0.9$, $\beta_2 = 0.999$
  - Loss Function: Dice + cross-entropy combined.

## 2 Transformer-based Prompt Encoder Architecture

The MaskTransformer uses $L = 3$ encoder blocks, hidden size $d_{\mathrm{model}} = 64$, and $H = 4$ self-attention heads. Table 1 shows the detailed parameter for the 3-layer MaskTransformer.

## 3 Prompt Injection into U-Net

During decoding we first obtain a multi-channel mask embedding $\mathbf{e} = \texttt{MaskTransformer}(\mathbf{m}) \in R^{B \times C \times 1 \times 1 \times 1}$ or a one-hot vector $\mathbf{o} \in R^{B \times C \times 1 \times 1 \times 1}$. A lightweight `PromptGenBlock` upsamples $\mathbf{e}$ or $\mathbf{o}$ to the current feature-map resolution and produces a prompt tensor $\mathbf{p} \in R^{B \times C \times D \times H \times W}$. We concatenate this prompt with the decoder feature map $\mathbf{f}$ *before* the first residual block at each resolution level:

$$\mathbf{f}' = \mathrm{concat}(\mathbf{f}, \mathbf{p}) \quad \in R^{B \times 2C \times D \times H \times W}.$$

The concatenated tensor passes through two *time-conditioned* ResNet blocks, a $1 \times 1 \times 1$ convolution for channel mixing, and finally merges with the corresponding skip feature from the encoder (Figure 4, Step 2). We perform this injection at three decoder resolutions ($\frac{1}{4}$, $\frac{1}{8}$, $\frac{1}{16}$).

Table 1: Layer shapes and parameter counts for the 3-layer MaskTransformer ($24^3$ input).

| Layer | Output shape | Kernel / Stride | Params (k) |
|-------|-------------|-----------------|------------|
| 3-D Patch Emb. | $B \times 64 \times 3 \times 3 \times 3$ | $8^3/8^3$ | 3 |
| Flatten $\rightarrow$ Tokens | $B \times 27 \times 64$ | — | — |
| *Transformer Encoder ($L$=3, $H$=4, FF=256)* | | | |
| Block 1 | $27 \times B \times 64$ | — | 43 |
| Block 2 | $27 \times B \times 64$ | — | 43 |
| Block 3 | $27 \times B \times 64$ | — | 43 |
| Global AvgPool | $B \times 64$ | — | 0 |
| MLP Head (LN+FC) | $B \times 64$ | — | 4 |
| **Total** | — | — | **136** |

## 4 Computational Cost

Table 2 summarises the hardware and runtime of each stage. All models were trained on a single *Tesla V100-SXM2-32GB*. The VQGAN pre-training is the most time-consuming step (168 h), while the diffusion baselines require roughly four days each. At inference, PromptLesion achieves $\approx 3.1$ volumes min$^{-1}$ for $96^3$ patches with 400 sampling time steps, comparable to Med-DDPM and only slightly slower than DiffTumor.

Table 2: Computational Details

| Model | GPUs | Train time | Vol/min |
|-------|------|-----------|---------|
| VQGAN | 1 x Tesla V100-SXM2-32GB | 168 h | - |
| PromptLesion | 1 x Tesla V100-SXM2-32GB | 96 h | 3.1 |
| DiffTumor | 1 x Tesla V100-SXM2-32GB | 96 h | 4.0 |
| Med-DDPM | 1 x Tesla V100-SXM2-32GB | 96 h | 3.3 |

## 5 Perceptual Metrics (3-D FID)

We compute a lesion-focused 3D FID using the same Frechet distance formula as in Sun et al. (2022), but with two key adaptations: (1) we first crop each volume to the lesion's 3D bounding box (plus a 16-voxel margin) to remove background, (2) we mask-pool original inputs so that only voxels inside the lesion contribute.

Table 3 lists the resulting FID↓ scores. DiffTumor, which is trained independently for every organ, achieves the best (lowest) value on every modality—exactly what one would expect given its training objective. PromptLesion comes second overall, ahead of Med-DDPM, which is consistent with its stronger generative prior but the absence of FID-specific fine-tuning. Note that, despite these FID rankings, PromptLesion still yields the highest segmentation Dice in our main results, reaffirming that lesion-FID and downstream task performance are only loosely correlated on high-contrast medical CT; we therefore report FID for completeness while using Dice to judge practical utility.

## 6 Synthetic Image Examples

We provide examples of synthetic lesions generated by the diffusion model on CCTA data for calcified and non-calcified plaques, as well as on CT scans for tumors in the liver, kidneys, and pancreas. The model inserts these lesions into anatomically consistent positions by conditioning on healthy patches, ensuring that the generated region blends naturally with the surrounding structures.

Table 3: 3-D FID ↓ metric

| Method | Calcified | Non-calcified | Liver | Kidney | Pancreas | Average |
|---|---|---|---|---|---|---|
| DiffTumor | 4.3 | 5.5 | 5.2 | 5.3 | 4.8 | **5.0** |
| Med-DDPM | 4.5 | 6.8 | 5.9 | 5.0 | 6.7 | 5.8 |
| PromptLesion | 5.5 | 5.9 | 4.8 | 5.2 | 6.2 | 5.5 |
| PromptLesion$^\dagger$ | 5.3 | 5.8 | 5.0 | 5.3 | 5.2 | 5.3 |

Figure 1 illustrates examples of calcified plaque (left block) and non-calcified plaque (right block) generation in CCTA slices, while Figure 2 demonstrates synthetic tumors in the kidney, liver, and pancreas. In each block of columns, the top row shows a healthy CT/CCTA patch, the second row shows the corresponding ground-truth label, the third row is the diffusion-based synthetic patch containing the newly introduced lesion, and the bottom row shows the synthetic lesion mask. Each color-coded column represents a different lesion type or organ region (e.g., calcified plaque versus non-calcified plaque in CCTA, or various organ tumors in CT).

For non-calcified plaque synthesis specifically (right block of Figure 1), the grayscale intensities of the generated lesion can appear quite similar to the surrounding vessel tissue, making it challenging to discern by visual inspection alone. Nonetheless, our PromptLesion-based diffusion successfully inserts subtle non-calcified plaques into the healthy context by the prompt. When we use these synthetic volumes (along with the associated lesion masks) to train segmentation networks, they help alleviate class imbalance and enable better detection of rare or visually subtle lesions.

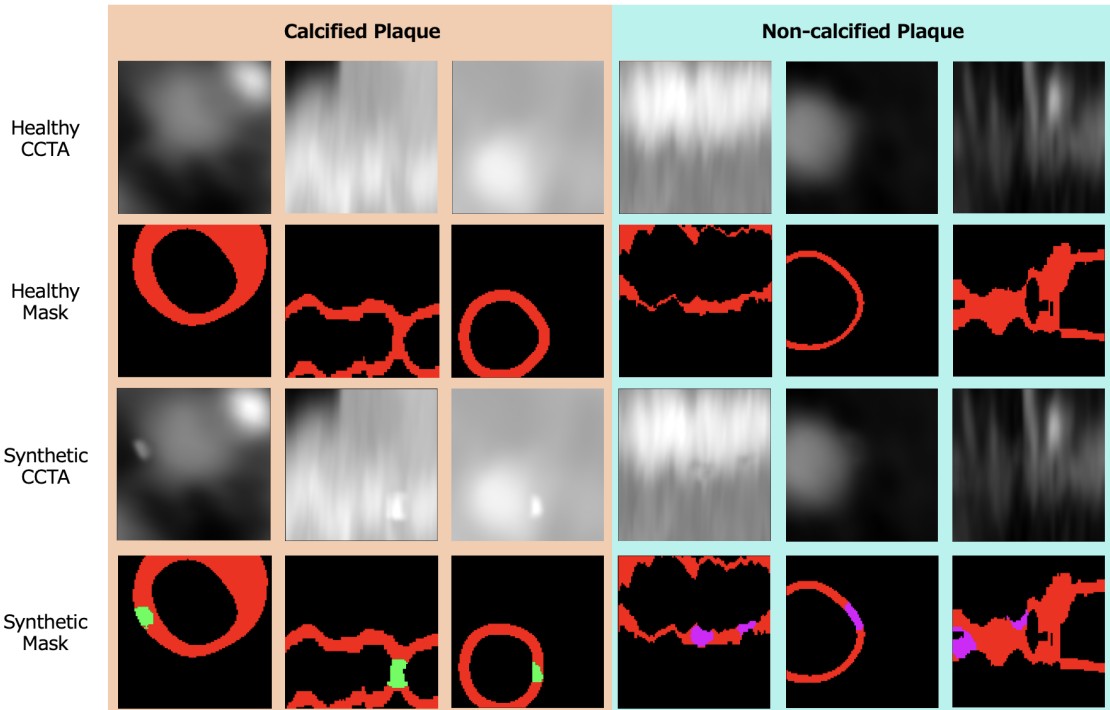

Figure 1: Synthetic images for CCTA datasets: In mask images, black refers to background, red denotes coronary artery wall, green indicates calcified plaque, and purple is non-calcified plaque.

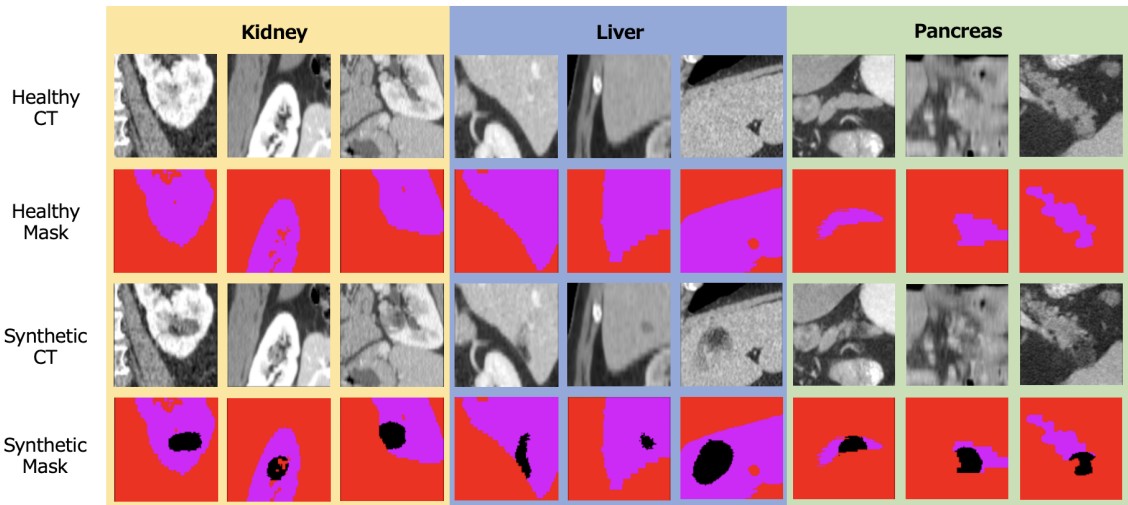

Figure 2: Synthetic images for tumour datasets: In mask images, red is the background, purple refers to organ, and black is the target tumour.

## 7    Pipeline

Figures 3–5 illustrate our end-to-end workflow:

- **Step 1 (Figure  3).** A VQGAN auto-encoder compresses the input volume into a discrete latent codebook.

- **Step 2 (Figure  4).** The PromptLesion diffusion model synthesises class-specific lesions in latent space, guided by either one-hot or transformer-based prompts.

- **Step 3 (Figure  5).** Synthetic lesions are decoded back to voxel space and mixed with real scans to train the segmentation network.

Together, these stages form the data-generation–to-segmentation loop used in all experiments.

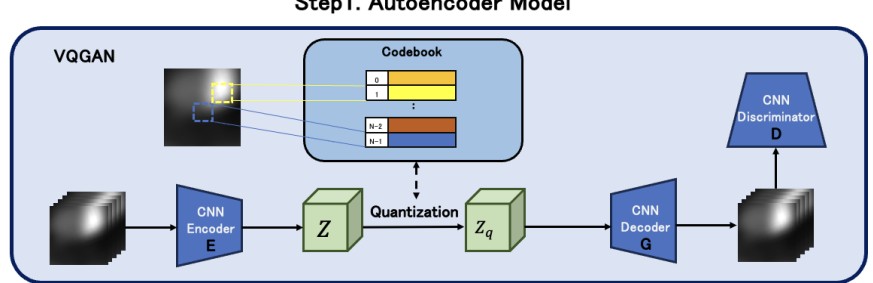

Figure 3: Overview for Step 1 (VQGAN autoencoder)

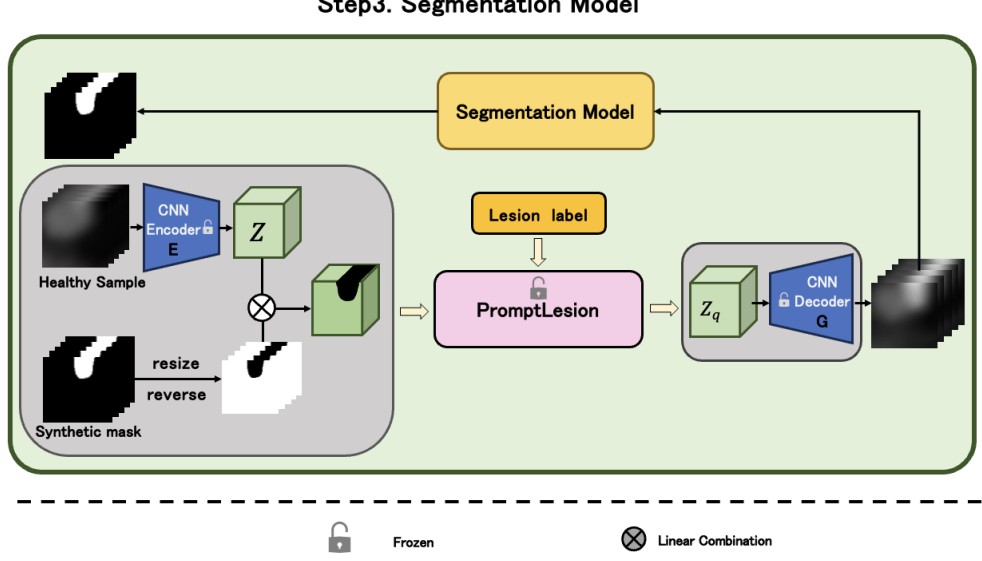

Figure 4: Overview for Step 2 (PromptLesion)

Figure 5: Overview for Step 3 (Segmentation)