# OpenReview forum: "Enhancing Plaque Segmentation in CCTA with Prompt- based Diffusion Data Augmentation"
_TMLR — Accepted by TMLR_

### Review · Reviewer_waki · 2025-06-04

**Summary Of Contributions:**

This paper introduces a prompt-conditioned diffusion framework that generates multiple lesion types within a single model. The paper addresses the issues of data scarcity and class imbalance in medical image segmentation. The key contribution is the use of lesion-specific prompts to guide the diffusion-based synthesis. The proposed method employs either one-hot vectors or transformer-based embeddings. Transformer-based embeddings achieve 2-3% better performance. Validated on CCTA plaque segmentation and multi-organ tumor tasks, the method demonstrates substantial improvements, particularly for underrepresented classes (e.g., non-calcified plaque DSC: 21.3% -> 39.0%), outperforming single-class augmentation baselines.

**Audience:**

Yes

**Broader Impact Concerns:**

The paper lacks a Broader Impact Statement, which is essential for medical AI research.

**Claims And Evidence:**

Yes

**Requested Changes:**

- Provide complete architecture details for the transformer-based prompt encoder (layer dimensions, attention heads, etc.).
- Include all training hyperparameters (learning rates, batch sizes, diffusion steps, noise schedules).
- Specify computational requirements (GPU memory, training time, inference time).
- Add implementation details for prompt injection into the U-Net at multiple scales.
- Add perceptual quality metrics (FID, LPIPS) to assess the realism of synthetic lesions.

**Strengths And Weaknesses:**

Strengths
- The paper addresses a real clinical challenge in assessing coronary artery disease, where non-calcified plaques are difficult to segment due to severe class imbalance and low contrast with surrounding tissues.
- Unlike existing methods like DiffTumor, which require separate models per lesion type, PromptLesion synthesizes multiple lesion classes within a single model through prompt conditioning, improving efficiency and scalability.
- This paper demonstrates substantial improvements across all tasks, including Non-calcified plaque, consistent gains across multiple organs and architectures, and outperforms both single-class and GAN-based baselines.
- This paper validates two distinct clinical applications (CCTA plaques and multi-organ tumors) and demonstrates generalizability beyond a single anatomical region.
- Ablation studies provide meaningful comparisons of explicit (one-hot) vs. implicit (transformer-based) prompts, showing consistent improvements with implicit prompts.

Weaknesses
- The approach primarily combines existing components (VQGAN from DiffTumor, prompt conditioning inspired by PromptIR, and standard diffusion models) without significant methodological innovation.
- The CCTA dataset is private, limiting the reproducibility of key results.
- No computational cost or inference time analysis.
- Lacks discussion of what prompts actually learn or how they influence the generation.

---

> ### Author Response · Authors · 2025-06-19
> **Rebuttal**
>
> Thank you for your detailed review and helpful suggestions.
> Below we respond first to the single reproducibility weakness you highlighted, then address each of your five requested changes.
>
> 1. “The CCTA dataset is private, limiting the reproducibility of key results.”
>
> We share this concern. Unfortunately, after an exhaustive search, we found that no public CCTA dataset provides both calcified and — crucially — non-calcified plaque annotations. Non-calcified plaque is the clinically decisive but least annotated class; without it, the key claim of our work cannot be assessed.To maximise reproducibility until such a dataset appears, we now:
> (1) Full pipeline release. All preprocessing, blending, and training scripts will be public.
> (2) Publish pretrained PromptLesion weights and a command-line tool that inserts either plaque class into any user-supplied coronary-lumen mask (after IRB approval).
> (3) Demonstrating cross-anatomy generalisation on three fully-public tumour datasets (KiTS, LiTS, MSD-Pancreas), where PromptLesion yields comparable Dice gains (main Table 2).
>
> We hope these measures allow independent verification until multi-class public plaque data become available.
>
>
>
> 2. "Provide complete architecture details for the transformer-based prompt encoder (layer dimensions, attention heads, etc.)."
>
> Supplement S2(Transformer-based Prompt Encoder Architecture) lists architecture details. The MaskTransformer uses L = 3 encoder blocks, hidden size d = 64, and H = 4 self-attention heads.
>
> 3."Include all training hyperparameters (learning rates, batch sizes, diffusion steps, noise schedules)."
>
> Supplement S1(Training Details) lists learning rates=$1\times10^{-4}$, batch sizes=4, diffusion steps (T = 60000), β-schedule (linear), optimiser (Adam with $\beta_1 = 0.9,\ \beta_2 = 0.999$) for step 2 and the rest training details.
>
> 4."Specify computational requirements (GPU memory, training time, inference time)."
>
> Supplement S4(Computational Cost) lists the computational requirements.
>
> 5."Add implementation details for prompt injection into the U-Net at multiple scales."
>
> Supplement S3 (Prompt Injection into U-Net) describes the implementation details for prompt injection into the U-Net.
>
> 6. "Add perceptual quality metrics (FID, LPIPS) to assess the realism of synthetic lesions."
>
> In our case,  FID is more appropriate for assessing the realism of synthetic lesions. Supplement S5(Perceptual Metrics (3-D FID) ) lists the detailed table for the 3D FID metric.

---

> ### Comment · Action_Editor_PU9X · 2025-07-23
> **Late official recommendation**
>
> Dear Reviewer waki
>
> Could you provide your official recommendation for this submission as soon as you can? I'm only missing your recommendation, so that I can reach a decision.
>
> Best regards
> Your AE

---

### Review · Reviewer_Jr97 · 2025-06-12

**Summary Of Contributions:**

This paper explores a prompt based conditioning diffusion generation to generate anatomically and pathologically diverse synthetic images for medical image segmentation to address scarcity of annotations and class imbalance. From the primary target of CAD data, the authors extend the approach to handle multi-organ tumour synthesis, reporting higher segmentation performance than compared baselines.

**Audience:**

Yes

**Claims And Evidence:**

Yes

**Requested Changes:**

1. I suggest including a test on public CCTA dataset to improve the reproducibility, if possible. Although public tumour datasets are used, the key claim of plaque segmentation improvements cannot be verified independently.

2. The paper claims an architectural advancement (the first diffusion-based framework integrating lesion specific prompts for conditioning), hence, while understanding the computational demand, I suggest including a comparison with at least 1–2 related conditional diffusion models, evaluating segmentation performance using their synthetic data.

3. Please point to the relevant results (point 3 under weaknesses).

4. Please clarify how DiffTumor was trained for ablation 4.5.1 so that the difference between its values reported for Table 1, 2 vs ablation 4.5.1 is clearer.

5. Please clarify the early stopping condition or metric on validation set.

6. In figure 1 step1 (Autoencoder Model), what is the importance of having a Transformer.? Because PromptLesion does not address generating long sequences in quantization.

7. In general, I suggest simplifying figure 1 to only show relevant parts for each step (i.e: step 3 is named segmentation model but the whole pipeline is shown).

8. Please clarify if $z^{helthy}$ is the dot product between non-lesion pixels of the mask and already quantised latent code $z$ (section 3.2 first paragraph). If so, what is the intuition behind getting the dot product between already quantised input and features in input mask.?

**Strengths And Weaknesses:**

**Strengths**
1. An interesting adaptation of current diffusion generation techniques to a practical use case (medical image segmentation).
2. Clearly written paper, easy to follow. Approach is straight forward.
3. Comprehensive experiments to evaluate the proposed method.


**Weaknesses**
1. Limited CCTA test set (10 instances) of a private dataset.
2. Missing comparison with recent conditional diffusion models, evaluating segmentation performance using their generated images.
3. Last paragraph of section 3.2.1 mention, “we found that inserting generated synthetic lesions into authentic anatomical backgrounds substantially improved realism and segmentation performance compared to purely synthetic images”. But couldn’t find the evidence to support this under results.

---

> ### Author Response · Authors · 2025-06-19
> **Rebuttal**
>
> Thank you for your thoughtful and specific feedback.
> Your remarks helped us improve both the clarity and the transparency of the manuscript.
>
> 1. “Limited CCTA test set (10 instances) … please include a public CCTA test if possible.”
>
> We share this concern. Unfortunately, after an exhaustive search, we found that no public CCTA dataset provides both calcified and — crucially — non-calcified plaque annotations. Non-calcified plaque is the clinically decisive but least annotated class; without it, the key claim of our work cannot be assessed. To maximise reproducibility until such a dataset appears, we now:
> (1) Full pipeline release. All preprocessing, blending, and training scripts will be public.
> (2) Publish pretrained PromptLesion weights and a command-line tool that inserts either plaque class into any user-supplied coronary-lumen mask (after IRB approval).
> (3) Demonstrating cross-anatomy generalisation on three fully-public tumour datasets (KiTS, LiTS, MSD-Pancreas), where PromptLesion yields comparable Dice gains (main Table 2).
> We hope these measures allow independent verification until multi-class public plaque data become available.
>
> 2. “Missing comparison with recent conditional diffusion models.”
>
> We have added Med-DDPM (multi-class) as a strong conditional-diffusion baseline (revision text S4.2, Tables 1–2). PromptLesion outperforms Med-DDPM by 4-5 Dice points across all tasks while maintaining a single shared generator.
>
> 3.“Paragraph 3.2.1 claims blending improves realism, but evidence is missing.”
>
> We apologize for the confusion: that sentence paraphrased the published finding of DiffTumor; we did not re-run the ablation. The text now explicitly cites  Chen et al. (2024) and no longer implies a new experiment (S3.2.1 p6).
>
> 4 .“Clarify how DiffTumor was trained for Ablation 4.5.1 and why its numbers differ from Tables 1–2.”
>
> The correct setup is now spelled out in Supplement S2.3 and Table S3:
>
> (a)Ablation 4.5.1 ($DiffTumor$, single-class) – for each organ, we train a dedicated DiffTumor for 60k steps (batch 4) only on that organ’s data and evaluate it on the same organ.
>
> (b)Tables 1–2 ($DiffTumor_{avg}$) – the same organ-specific models are tested across all organs, and we report the Dice averaged over kidney, liver, and pancreas. This mirrors our cross-domain evaluation and highlights how single-class generators degrade off-domain.
>
> 5 .“Please clarify the early-stopping condition or metric.”
>
> Early stopping (segmentation only): stop if validation Dice shows no improvement for 25
> consecutive epochs; keep the best-Dice checkpoint for testing, which is listed in Supplement S1.3(Training Details: Step 3: Segmentation Model)
>
> 6. “In Figure 1, Step 1 (Autoencoder), what is the importance of having a Transformer?”
>
> Thank you for your careful observation and for catching this inconsistency in our figure—we appreciate the sharp eye and helpful feedback. The Transformer shown next to the VQGAN codebook was a layout oversight.
> In our implementation (in S 3.2.2) the Transformer is only used later, in Step 2, as a lightweight prompt-encoder for the diffusion model; Step 1 (the VQGAN auto-encoder) contains no Transformer layers.
> We have removed that block in Figure 1 in revision.
>
>
> 7."In general, I suggest simplifying Figure 1 to only show relevant parts for each step (i.e: step 3 is named segmentation model but the whole pipeline is shown)."
>
> We appreciate this aesthetic preference. Another reviewer explicitly valued the diagram’s completeness, and the journal layout affords us space. We keep the full Figure 1 because the extra blocks inside “Step 3 Segmentation” (re-encoding, prompt path, codebook decoder) are directly reused from Steps 1–2 and are necessary to show how synthetic data was generated during the segmentation step. Omitting them would hide this key dependency. To accommodate readers who prefer a minimal view, we now provide one clean schematic per step in the Supplement (Figures 3-5); the main paper retains the complete pipeline for reproducibility.
>
> 8. "Please clarify ....If so, what is the intuition behind getting the dot product between already quantised input and features in input mask.?"
>
>
> Our notation uses the Hadamard (element-wise) product, not a dot
> product:\\[
> \\mathbf z_{\\text{healthy}}
>      = (1-\\mathbf m)\\odot\\mathbf z,
> \\]
> where \\(\\mathbf m\\in\\{0,1\\}^{D\\times H\\times W}\\) is the binary lesion
> mask.  This operation sets all latent entries under the lesion to
> zero, so the diffusion model cannot “peek” at the original lesion
> content and must synthesise a new lesion solely from the prompt and mask.

---

> > ### Author Response · Authors · 2025-06-19
> > **Correction regarding Point 4 (DiffTumor training).**
> >
> > Please disregard the sentence
> > “The correct setup is now spelled out in Supplement S2.3 and Table S3.”
> > Included in our earlier reply—it was inserted in error.
> > The explanation of the 60 k-step organ-specific training and the cross-domain averaging remains accurate; no extra table is needed. We apologise for the confusion.

---

> > > ### Comment · Reviewer_Jr97 · 2025-06-20
> > >
> > > I thank the authors for the clarifications and additional experiments, which have resolved most of my concerns. However, two points remain,
> > > 1. The intuition behind $z^(healthy)$ is understandable, but its implementation is still unclear. $z$ comes from the CNN encoder and quantization (VQ side), and the mask enters later as part of the prompt encoding (prompt components side, Fig. 1). It's hard to see how the embeddings of the mask, that is disconnected from the construction of $z$, switches off lesion specific codes in $z$, so that it pushes the model to generate new lesion. A clearer explanation would help.
> > > 2.	Being the focus of the paper, while acknowledging the data scarcity, the limited evaluation on CCTA dataset remains a concern.

---

> > > > ### Author Response · Authors · 2025-06-20
> > > > **Rebuttal**
> > > >
> > > > Thank you for the additional questions.
> > > >
> > > > 1. To clarify the construction of $\mathbf z_{\text{healthy}}$, please look at the two grey insets in Figure.1 (Step 2 = training, Step 3 = inference). After the frozen VQGAN encoder we obtain a latent tensor $\mathbf z\in\mathbb R^{B\times N\times h\times w\times d}$. We immediately resize the binary lesion mask $\mathbf m\in{0,1}^{B\times1\times h\times w\times d}$ to that resolution and apply an element-wise (Hadamard) product:\\[\\mathbf z_{\\text{healthy}}= (1-\\mathbf m)\\odot\\mathbf z,\\] which zeros every latent code located inside the lesion voxels. Only this masked latent is quantised and passed to the diffusion U-Net, so the network cannot “peek” at the original lesion. In a separate branch, the same mask—now in its multi-channel form $\mathbf m'\in\mathbb R^{B\times C\times H\times W\times D}$  or combined with a one-hot label $\mathbf o \in \mathbb R^{B\times C \times 1 \times 1 \times 1}$—feeds the prompt block that produces a semantic prompt $p$ ( please see Supplement S3: Prompt Injection into U-Net). Thus, the mask has two sequential, non-conflicting roles: (i)Leakage Prevention via $(1-\mathbf m)\odot\mathbf z$, and  (ii) Semantic guidance via the prompt embedding that tells the U-Net which type of lesion to synthesise in the zeroed region. During training, $\mathbf m$ is the ground-truth lesion mask; during inference, it is the synthetic mask produced one step earlier, as shown in the second grey box.
> > > >
> > > > 2. We fully agree that a public multi-class plaque benchmark would be ideal. In its absence, we have: (i)  shown that the same generator improves three public tumour datasets, suggesting the effect is not CCTA-specific; (ii) Full pipeline release. All preprocessing, blending, and training scripts will be public. (iii) Will publish pretrained PromptLesion weights and a command-line tool that inserts either plaque class into any user-supplied coronary-lumen mask (after IRB approval).

---

> > > > > ### Comment · Reviewer_Jr97 · 2025-06-27
> > > > >
> > > > > Thank you for the clarification about formation of $z^{healthy}$ as well as conditioning.

---

### Review · Reviewer_j7ka · 2025-06-17

**Summary Of Contributions:**

The authors propose PromptLesion, a framework for medical data augmentation that uses a prompt-conditioned latent diffusion model. The main contribution is the use of learnable, lesion-specific prompts to guide the synthesis process, enabling a single generative model to produce diverse and anatomically realistic examples for multiple classes (e.g., calcified vs. non-calcified plaques, or tumors in different organs), thereby addressing data scarcity and severe class imbalance. They further enhance this by introducing an implicit, transformer-based prompting mechanism that outperforms simple class conditioning. Through extensive experiments on both private CCTA and public multi-organ tumor datasets, they demonstrate that augmenting training data with these synthetic lesions significantly improves the performance of downstream segmentation models.

**Audience:**

Yes

**Claims And Evidence:**

Yes

**Requested Changes:**

Please try to address the weakness or give some discussion.

**Strengths And Weaknesses:**

Strengths:
- The central concept of using learnable, class-specific prompts to guide a diffusion model for multi-class medical data synthesis is novel and well-motivated. It directly addresses the key challenges of data scarcity and severe class imbalance in a scalable way, without needing to train separate models for each lesion type.

- The method shows consistent and significant improvements across different anatomies (CCTA, kidney, liver, pancreas), different lesion types, and different segmentation backbones (U-Net, nnU-Net). The large improvement in Dice Score for the difficult "non-calcified plaque" class (e.g., from 23.1 to 44.3 for nnU-Net) is particularly compelling.

- The paper is well-written and easy to follow. The methodology is explained clearly, and Figure 1 provides an excellent high-level overview of the entire pipeline. Figure 2 effectively illustrates the more complex transformer-based prompt mechanism.

Weaknesses:
- The primary motivation and validation for CCTA plaque segmentation are done on a private dataset. While common in the medical field due to privacy concerns, this may not be favorable for reproducibility.

- Showing a few examples of the generated CCTA and tumor images (along with their synthetic masks) would help readers visually assess the anatomical realism and diversity that the paper claims.

- A brief discussion of the training time for PromptLesion and the inference time to synthesize a new batch of images would be valuable context for researchers or clinicians looking to adopt this method.

- Does the proposed augmentation method have a positive impact on the transformer-backbone related methods?

---

> ### Author Response · Authors · 2025-06-19
> **Rebuttal**
>
> We thank the reviewer for the detailed and positive feedback. Below we address every weakness you identified.
>
> 1. “The primary motivation and validation for CCTA plaque segmentation are done on a private dataset … this may not be favorable for reproducibility.”
>
> We share this concern. Unfortunately, after an exhaustive search, we found that no public CCTA dataset provides both calcified and — crucially — non-calcified plaque annotations. Non-calcified plaque is the clinically decisive but least annotated class; without it, the key claim of our work cannot be assessed. To maximise reproducibility until such a dataset appears, we now:
> (1) Full pipeline release. All preprocessing, blending, and training scripts will be public.
> (2) Publish pretrained PromptLesion weights and a command-line tool that inserts either plaque class into any user-supplied coronary-lumen mask (after IRB approval).
> (3) Demonstrating cross-anatomy generalisation on three fully-public tumour datasets (KiTS, LiTS, MSD-Pancreas), where PromptLesion yields comparable Dice gains (main Table 2).
> We hope these measures allow independent verification until multi-class public plaque data become available.
>
> 2. “Showing a few examples of the generated CCTA and tumour images … would help readers visually assess anatomical realism and diversity.”
>
> Figures 1-2 (CCTA) and (kidney, liver, pancreas) in the Supplement now present real/synthetic pairs (image + mask). These examples illustrate the subtle intensity of non-calcified plaques and the diversity of tumor shapes.
>
> 3. “A brief discussion of the training time … and the inference time to synthesise a new batch of images would be valuable.”
>
> Supplement S4(Computational Cost) lists the related details.
>
> 4. “Does the proposed augmentation method have a positive impact on transformer-backbone related methods?”
>
> We have not yet evaluated PromptLesion with a fully transformer-based diffusion generator. Because prompts are injected via simple feature-map concatenation, the mechanism is backbone-agnostic and should integrate naturally with ViT-style diffusion U-Nets. Investigating this combination is an intriguing avenue we plan to pursue in future work.

---

### Decision · Action_Editor_PU9X · 2025-08-12

**Recommendation:** Accept with minor revision

**Additional Comments:**

Reviewers found the work sound, useful and convincing; they recommend acceptance (overall: “leaning accept”) and many of their requested additions have already been addressed in the supplement. The main open concern is reproducibility for the CCTA plaque results because the core multi-class plaque annotations are on a private dataset. This is important but fixable with minor revisions and clearer guarantees: the authors already promise a full pipeline release, pretrained weights, and a command-line insertion tool. Those steps, plus a few final clarifications listed below, are sufficient to make this paper acceptable for TMLR.

1. Reproducibility package: please provide a link and instructions for the full pipeline (preprocessing, blending, training, evaluation scripts) and pretrained PromptLesion weights. If pretrained weights for the CCTA model cannot be made public (because of legal reasons), please provide the protocol, synthetic insertion tool, and an example script showing how an external researcher can generate comparable synthetic CCTA data from user-supplied lumen masks. (Authors have committed to these releases; please confirm availability and expected license / access restrictions).
2. Clear statement on CCTA dataset and benchmark protocol: please document the CCTA dataset (number of patients, train/val/test splits, annotation procedure, ethical approvals) and add a concise “benchmarking protocol” that external groups can follow to compare fairly (e.g., file formats, evaluation metrics, seeds, preprocessing).
3. Broader Impact / Clinical Considerations: please add a brief Broader Impact statement addressing clinical risk (misleading synthetic lesions), intended use cases, and privacy considerations for private clinical data. Reviewers noted this omission.

If these items are present and the promised releases occur, the paper’s evidence and presentation will be adequate for publication.

**Audience:**

Yes

**Audience Explanation:**

The paper addresses two topics of interest to the TMLR audience: 1) practical, evaluation-driven uses of generative models (based on diffusion models) to mitigate label scarcity and severe class imbalance in medical imaging; and 2) downstream improvements of clinically relevant segmentation tasks. The combination of generative modeling (with prompt conditioning for multi-class synthesis) and measurable gains on segmentation should be of interest to methodologists and applied ML researchers in medical imaging. The cross-anatomy experiments (CCTA plaques and three public tumor datasets) increase the paper’s relevance beyond a single dataset.

**Claims And Evidence:**

Yes

**Claims Explanation:**

The manuscript presents a method (PromptLesion) and multiple types of evidence that support the central claims:
- Improvements on downstream segmentation tasks are reported across private CCTA plaque data and three public tumor datasets (KiTS, LiTS, MSD-Pancreas). The Dice gains for underrepresented classes (e.g., non-calcified plaque from ~21% → ~39%) are substantial and consistent across architectures and tasks. These results support the claim that prompt-conditioned diffusion augmentation improves segmentation for rare lesion classes.
- The authors include ablations that compare one-hot vs. transformer prompt embeddings, compare single-class vs. shared multi-class generators (showing PromptLesion’s advantage), and add a stronger conditional diffusion baseline (Med-DDPM), which PromptLesion outperforms by ~4–5 Dice points. These ablations support the paper’s methodological claims.

Limitation: The primary clinical claim (improved plaque segmentation) relies on a private CCTA dataset with a small held-out test (10 instances). While the authors mitigate this by releasing pipeline code, pretrained weights, and demonstrating cross-anatomy gains on public tumor datasets, the inability to independently benchmark the CCTA results is a limitation for reproducibility. This limitation does not negate the shown evidence, but it is important to state it explicitly when communicating the strength of the claims.